# Psychometric properties of the polish version of the Dysfunctional Thoughts about Caregiving Questionnaire (DTCQ)

**Katarzyna Sanna**[1☉*], **Maciej Kościelniak**[2☉], **Jarosław M. Michałowski**[3‡], **Andrés Losada-Baltar**[4‡], **Konrad Piotrowski**[1☉]

**1** Center for Research on Personality Development, Department of Psychology, SWPS University, Poznań, Poland, **2** Department of Psychology, SWPS University, Poznań, Poland, **3** Poznan Laboratory of Affective Neuroscience, Department of Psychology, SWPS University, Poznań, Poland, **4** Department of Psychology, Universidad Rey Juan Carlos, Madrid, Spain

☉ These authors contributed equally to this work.
‡ These authors also contributed equally to this work.
* ksanna@swps.edu.pl

## Abstract

Dysfunctional thoughts about caregiving are defined as rigid and inappropriate interpretations of one's experiences, behaviors, thoughts, or feelings that are associated with providing care to a dependent relative. The present study examined the psychometric properties of the Polish adaptation of the Dysfunctional Thoughts about Caregiving Questionnaire (DTCQ). DTCQ was administered to 618 family caregivers of people with chronic health problems. As the Confirmatory factor analysis supported both one and two-factor structure of the DTCQ, following the principle of parsimony, one-factor model was considered more appropriate. Construct validity was also supported by significant and theoretically expected relationships between DTCQ and depression, identity formation or sociodemographic variables. In conclusion, this study is the first to provide preliminary evidence for the reliability and validity of the DTCQ as a measure of dysfunctional thoughts in Polish family caregivers.

## Introduction

As caregiving for a chronically ill person has proved to be a stressful and burdening experience for many family members, an increasing need for an in-depth look at the factors affecting the health and well-being of family caregivers has emerged [1–4]. In the literature, family members are called family caregivers, informal carers, family carers, or closest relatives who nurse their spouses, children, parents, distant relatives, or unrelated people [5]. Family caregivers perform a large number of tasks for many years and many hours per day that are associated with the illness, household duties, or emotional support of patients or other relatives [6–7]. As most family caregivers are not professionally prepared to perform specialized care-related activities,

**Data availability statement:** All files will are available from the Open science framework: https://osf.io/r94q2/?view_only=4c3f-174962674c2e8b4570e5c82c63d5

**Funding:** The authors Katarzyna Sanna, Konrad Piotrowski, Maciej Kościelniak, Jarosław Michałowski received a research grant as part of the 6th call for grant proposals for financing research projects at the Institute of Psychology (6 konkurs na finansowanie projektów badawczych w Instytucie Psychologii), SWPS University, Warsaw, Poland (Grant number - 1268-11). The funders had no role in study design, data collection and analysis, decision to publish, or preparation of the manuscript.

they are at risk of an increased sense of distress and burden, followed by other mental health problems, i.e., depression or anxiety [8–14]. Due to the challenges of an ageing society, increases in life expectancy even after being diagnosed with a life-threatening disease and the availability of institutional support, the adherence to personal values related to the importance of family (e.g., familism), and being cared for at home, family caregivers may remain the main source of care for many people [15]. Thus, the well-being of this population should be a significant issue for health-care providers and researchers studying the impact of health problems on family system functioning.

One of the factors related to the quality of life of family caregivers is their personal beliefs about 'good care' and their self-image as a caregiver. Unfortunately, the experience of caring for a dependent relative is often associated with the development and activation of many dysfunctional thoughts related to the care, which are defined as rigid and involve inappropriate interpretations of one's experiences, behaviors, thoughts, or feelings [16–17]. These types of thoughts that are learned through life experiences (e.g., influence of the family or the cultural context) impose rigid, inflexible, and maladaptive rules of coping with caregiving (e.g., "A caregiver should only seek help from others or find other alternatives when the caregiving situation is at its worst or when they can no longer handle it") and guide behavior (e.g., not seeking help and not devoting time to rest or leisure) [18]. These ways of thinking can act as cognitive barriers that make it difficult for the caregivers to take care of their own mental and physical health. Specifically, Montorio et al. [17] identified three types of interrelated dysfunctional beliefs connected to the care-giving experience. First, some of these beliefs may affect the amount of help that the caregivers receive for themselves. For instance, a caregiver may not seek any assistance from other family members or professionals as a result of a belief that a good caregiver should be able to handle all the caregiving tasks alone. Second, dysfunctional beliefs may create an excessive sense of perfectionism, responsibility, or excessive involvement in care, as the caregiver believes that only they are able to perform all care-related demands best. Third, this rigid way of thinking may prevent the expression of difficult emotions, as some caregivers believe they should not complain or verbalize any negative feelings. As a consequence, the caregiver does not allow themself to talk about the difficulties of care, which in the long run may result in a greater sense of burden. This way of thinking is highly dependent on cultural context, however. In particular, cultural values like familism are considered a factor that influences the way family members perceive their tasks regarding caregiving [20]. In cultures that cultivate family traditions, including Polish society [21], there is a cultural message that indicates how caregivers should respond to the needs of dependent relatives, and that responsibility for care lies within the family [22]. According to the cognitive-behavioral model, these thoughts are responsible for the appearance of a negative mood, and in the long run for, among other effects, depression or anxiety [23–24].

Various dysfunctional thoughts-related questionnaires have been used in caregiving research. All of them were developed in the context of studies focused on family

caregivers of people with dementia [19,25,26]. One of the existing scales is the Dysfunctional Thoughts About Caregiving Questionnaire (DTCQ; [19]) —a self-report measure that assesses two different domains of caregiving [17]. The first domain refers to "perception of sole responsibility" in which the caregiver believes that a "good carer" should put care first and subordinate their own well-being and interests to the caregiving. The second domain refers to a sense of perfectionism, which represents high levels of emotional and behavioral self-demands. The questionnaire has high internal consistency and moderate test-retest reliability. The measure has Spanish, English, and French language versions [17,19,27], however the factor structure of the scale was verified only for the Spanish and English versions, showing the presence of two related factors. As the DTCQ theoretical framework relies on cognitive-behavioral model, and has already been tested in interventions in different settings, the authors of this study found this questionnaire to be appropriate in both research and practical contexts.

Although the DTCQ was developed with a sample of family caregivers of people with dementia, the informal care experience affects family members of people facing a variety of conditions [28–29]. To more closely investigate this issue, in the present study we decided to use the DTCQ in a new area and assess its validity in a study of caregivers of people with a variety of health-related problems, including dementia, somatic diseases, mental health problems, or old-age frailty.

We specifically investigated the situation of family caregivers in Poland. Thus, our main objective was first to adapt the DTCQ [19], which would allow for the assessment of dysfunctional beliefs of family caregivers in Poland. In order to further verify the search for evidence of validity, our second goal was to examine the relationships between the characteristics of family caregivers, contextual factors, and caregivers' quality of life.

Studies on the consequences of informal caregiving have sought to identify factors that may affect care-related burden and negative caregiving outcomes (e.g., depressive symptoms; [30–31]. Some previous research indicated that the level of social support is one of the agents that affects adaptation to the informal caregiving experience [32], which can also be related to dysfunctional thoughts about caregiving. It was recently shown, however, that it is the support from family members rather than formal, institutional, or social support that can mitigate most of the negative consequences of caregiving [33]. In turn, the demographic variable that was previously considered as most related to caregivers' health was the gender of the caregiver, with women, compared to men, being more burdened with care [34]. However, recent research indicated that men also suffer negative consequences from providing care [35].

Another characteristic possibly related to adaptation to caregiving is the formation of and compliance with the caregivers' identity [36]. It is possible that caregivers' thoughts or beliefs may participate in the formation of identity. Studies based on processual identity models have confirmed that having a stable identity in various domains (e.g., romantic relationship, parenting, vocational activity) is one of the key predictors of adaptation, coping, and a high quality of life [37]. To our knowledge, there were currently no studies on the formation of caregivers' identity among family caregivers. However, research regarding parental identity (also related to the care of others) has shown that a stable sense of identity is associated with greater generativity and higher psychological and social well-being [38]. On the other hand, poor identification with the parental role has been associated with a high intensity of depressive symptoms, parental burnout, and a low quality of life [39]. In the present study, we hypothesized that the development of the carer's identity can also be linked to the level of their dysfunctional thoughts about caring.

The present study involved the Polish adaptation translation and validation of the DTCQ among a diverse population of family caregivers. We also aimed to verify whether the DTCQ used in more diverse groups of caregivers would retain the psychometric properties observed in samples of family caregivers of people with dementia. Specifically, we expected that the Polish version of the DTCQ would be characterized by a two-factor structure, similar to the original version, and would have acceptable internal consistency over.70 [40]. Additionally, we assumed that the correlation between the two subscales (perception of sole responsibility and perfectionism) would be small to moderate, supporting two distinct aspects of dysfunctional beliefs related to caregiving. We also predicted that family caregivers' scores on the DTCQ dimensions would be related to the sense of identity in the caregiver. We predicted that a strong identification with the role

of caregiver, when "being a caregiver" is the centerpiece of identity around which only the rest of life's activities are organized, could be associated with a sense of sole responsibility and striving for perfectionism in caregiving. In terms of the identity model we used Croectti's theory [36], this could mean a fairly strong positive relationship between dysfunctional thoughts with identity commitment (identifying with the role of a caregiver and defining one's identity through this lens) and in-depth exploration (actively seeking information about how to fulfill the role of a caregiver, reflecting on and analyzing how an individual carries out the role of a parent), and a negative one with reconsideration of commitment, which means reluctance to take on the role of a caregiver. We also predicted the relationship between dysfunctional thoughts about care and contextual variables that influence caregiving (e.g., kinship relationship – where we assumed that the closer the relationship is, the more dysfunctional thoughts on the caregiver, type of illness of the relative – where we assumed that there would be no differences between the types and caregivers' dysfunctional thoughts level). Furthermore, we expected that DTCQ scores would positively correlate with depressive symptoms and negatively correlate with the social support the caregiver receives, and that higher levels of dysfunctional thoughts would be related to either a well-formed but rigid identity or to identity diffusion, manifested by a lack of acceptance of the caregiver role [39]. Regarding contextual factors, we predicted that the severity of dysfunctional thoughts would be positively related to the closeness of the relationship (i.e., the closer the relationship, the greater the sense of responsibility and perfectionism). As the DTCQ had been studied solely with dementia populations, we made no prior assumptions considering its relationships with the type of illness the close relative suffered from.

## Materials and methods

### Participants and procedure

The participants in this study were Polish family caregivers of people with chronic health problems. In order to participate in this study, respondents had to comply with the following inclusion criteria: caring for a sick family member/close relative and aged between 18 and 75 years. Before recruitment, we estimated the required sample size with G∗Power 3.1.9.6 software [41]. For pairwise correlation analysis, assuming an effect size of.25, an alpha level of.05, and a power level of.95, the minimum required sample size was 202. For a one-way ANOVA, with the three comparison groups (effect size.25, alpha.05, power.95) that we planned to use, the minimum sample had to be 324. Because the structure of the DTCQ had not previously been studied in Poland, we assumed that the results might not support our hypotheses. In order to be able to carry out more complex analyses in such a scenario, we considered recruiting a sample twice as large as the minimum.

This study was conducted with an online survey from 1st August 2022 to 22 November 2022, through SW Research, a professional Polish research and polling agency, which provides the opportunity to conduct research among registered panellists, whose structure corresponds to a miniature of Polish society. After the database of questionnaires was uploaded, it was published to respondents, who, after learning the purpose of the survey, could participate in it. Using their established participant panel, SW Research identified and recruited family caregivers who met our inclusion criteria: caring for a sick family member/close relative and aged between 18 and 75 years. The agency's sampling methodology ensured participant verification and data quality through their established panel management procedures.

In addition to the SW panel, i nformation about the study was published on public social media and promoted by students at the authors' university. Students were informed of the purpose of the survey and asked to send a link to the panel to potential respondents or support groups for sick people. For helping with recruitment, students received university credits. Written informed consent was obtained from all the participants enrolled in the research. The information included a statement about the purpose of the project and that participation was voluntary and free to terminate at any time; research specification including the number of questionnaires and time needed to complete them, the costs and benefits of participating in the project, and the contact information of the main investigator in case of any questions. We excluded people who did not confirm that they were caring for a dependent person and people younger than 18 years and those

older than 75 years. Once a participant selected all the required checkboxes, they were directed to the research survey questionnaires.

Six hundred and eighteen caregivers of people chronically ill or requiring care participated in this study (*n* = 479; 77.5% female). As shown in Table 1, most of the participants were caring for a frail older adult, followed by care-recipients with a somatic disease, mental health disease, and dementia. Most family caregivers cared for a parent (*n* = 271; 43.7%) or a grandparent (*n* = 182; 29.5%). The remaining sociodemographic characteristics of the sample are shown in Table 1.

**Table 1. The descriptive characteristics of family caregivers in this study's sample.**

| Characteristic | |
|---|---|
| Gender | Female: n = 479, 77,5%<br>Male n = 139, 22,5% |
| Age | M = 42.2<br>SD = 14.84<br>Range = 18–75 |
| Education (%) | Higher education: n = 304, 49.2%<br>Secondary education: n = 235, 38.2%<br>Primary education: n = 12, 1.9%<br>Vocational education: n = 61, 9.9%<br>Basic education: n = 5, 0.8% |
| Marital status (%) | Formal relationship: n = 300, 48.5%<br>Informal relationship: n = 175, 28.3%<br>Single: n = 143, 23.1% |
| Living area (%) | Countryside: n = 144, 23.3%<br>City > 10,000: n = 103, 16.7%<br>City > 100,000: n = 145, 23.5%<br>City > 500,000: n = 112, 18.1%<br>City < 500,000: n = 114, 18.4% |
| Kinship (per the person needing care) (%) | Spouse: n = 58, 9.4%<br>Parent: n = 271, 43.9%<br>Child: n = 54, 8.7%<br>Sibling: n = 25, 4%<br>Cousin: n = 2, 0.3%<br>Grandpa/grandma: n = 182, 29.4%<br>Unrelated friend: n = 26, 4.2% |
| Caregiving reasons (%) | Somatic disease: n = 146, 23.6%<br>Mental health disorder: n = 80, 12.9%<br>Old-age frailty: n = 324, 52.4%<br>Dementia: n = 68, 11.1% |
| Level of the caregiver's perceived limitation of the caregiving (%) | Not at all: n = 21, 3.4%<br>To a small extent: n = 149, 24.1%<br>To a medium extent: n = 253, 40.9%<br>Mostly: n = 169, 27.3%<br>Entirely: n = 26, 4.2% |
| Financial situation (%) | Bad: n = 56, 9.1%<br>Average: n = 336, 54.4%<br>Good: n = 180, 29.1%<br>Very good: n = 46, 7.4% |
| Perceived health of the caregiver | Bad: n = 28, 4.5%<br>Average: n = 247, 40%<br>Good: n = 268, 43.4%<br>Very good: n = 75, 12.1% |

## Measures

**The Dysfunctional Thoughts About Caregiving Questionnaire (DTCQ).** The DTCQ [19] consists of 16 items and examines two dimensions: perception of sole responsibility (e.g., "*No matter how badly a caregiver feels, he/she should not vent with others because it would be disrespectful to the person being cared for*"), which means that the caregiver's priorities are the needs of the person under care, and perfectionism (e.g., "*A good caregiver should never get mad or lose control with the person that is being cared for*"), which represents a high level of caregiving demands in both emotional and behavioral aspects. All items were assessed on a 5-point Likert scale. All items were assessed on a 5-point Likert scale, from 0 (I completely disagree) to 4 (I completely agree). Information on the factor structure and reliability of the scale is given in the following section.

With the consent of the authors of the original DTCQ [19], a standard back-translation was performed [42]. The questionnaire was translated into Polish by three independent translators. The fourth judge then compared the resulting versions, and the final Polish version of all items was determined. A native English speaker later translated the questionnaire back into English and compared it with the first author's original version. The Polish version prepared in this way was used in the present study. The psychometric properties of the Polish version are described in the results section below.

**The Symptom Checklist-27-Plus (SCL-27-Plus).** The SCL-27-plus [43] is a short, multidimensional screening instrument for mental health problems, containing five scales: depressive, vegetative, agoraphobic, social phobia, and pain symptoms. Each scale consists of between four and six items. The results are assessed based on the average of these items, with a higher score indicating a higher severity of symptoms of mental health problems. In this study, only a scale measuring the severity of depressive symptoms was used, consisting of five items, assessed based on their frequency of occurrence (the scale ranged from 1 (*never*) to 5 (*very often*). Cronbach's alpha was calculated, showing good reliability (α = 0.88).

**The Multidimensional Scale of Perceived Social Support (MSPSS).** The MSPSS [44] is a self-report measure to assess perceived social support. It consists of 12 items with a 7-point Likert scale, from 1 (*I completely disagree*) to 7 (*I completely agree*). The MSPSS has a three-factor structure that refers to different sources of social support from family members, friends, or significant others. The results can be calculated both for individual subscales and for the entire scale. The higher the reported score, the higher the level of social support perceived by the person. Cronbach's alpha was calculated, showing excellent reliability (α = 0.93) for the Total score, Friends subscale (α = 0.93), Family subscale (α = 0.93) and Significant Others subscale (α = 0.92).

**The Utrecht-Management of Identity Commitments Scale, Informal Care (U-MICS-IC).** The U-MICS [36,45] measures the processes involved in identity development in different life domains. For this study, the scale was adapted for research with family caregivers (U-MICS-IC) by slightly modifying the items to relate to the care of a close adult. The scale consists of 13 items and assesses three processes: (1) identity commitment (e.g., "*Being a caregiver [of this person] gives me a sense of security in life*"), (2) in-depth exploration (e.g., "*I often think about caring for [this person]*"), and (3) reconsideration of commitment (e.g., "*I often think that not having [that person] would make my life more interesting*"). A 5-point Likert scale is used to assess the individual items, with 1 (*completely false*) to 5 (*completely true*) answers. Since this was the first use of this version of the questionnaire, we conducted confirmatory factor analysis (CFA) to evaluate its hypothesized three-factor structure. The analysis was performed with AMOS 29 [46] and our results, similarly to the original version [36,45], confirmed the theoretical validity of the scale with the three-factor solution fitting the data well(CFI =.95, SRMR =.05, and RMSEA =.07) [47]. Cronbach's alpha was calculated, showing excellent reliability for Identity commitment dimension (α = 0.90, acceptable reliability for In-depth exploration dimension (α = 0.79) and Reconsideration of commitment dimension (α = 0.78).)

**Sociodemographic characteristics.** A short sociodemographic questionnaire was created that included gender, marital status, level of education, reasons of caregiving, the level of perceived limitation from the caregiving, perceived health of the caregiver, and financial situation. Living area was classified into 5 levels, with a higher number indicating a

locality with a greater number of inhabitants. The relationship with the person needing care was evaluated with a 7-point list of alternatives (i.e., "*he/she is my child*," "*he/she is my parent*," "*he/she is my friend, we are not related*")

### Ethical considerations

This study received approval from the research ethics committee at the authors' university (Komisja ds. Etyki badań naukowych w Poznaniu, SWPS; Commission on Research Ethics in Poznan, SWPS) (reference no. 2022–143, July 12, 2022).

### Data analyses

We performed statistical analyses with JASP [48] (Version 0.14.1), SPSS (Version 29.0.2.0) [49], and Mplus [50]. In the first step, we conducted a normality analysis of all the studied continuous variables. To assess their univariate normality, we applied the cutoff criteria for skewness and kurtosis between -2 and +2 [51]. The distribution of the continuous variables turned out to be symmetrical, apart from perceived social support (MSPSS).In order to analyze the factorial validity of the DTCQ, we compared one- and two-factor solutions [17,19] with CFA. The analysis was performed with Mplus software [50], with the robust maximum likelihood (MLR) estimator. We applied several commonly used fit indices to evaluate and compare both solutions: comparative fit index (CFI), root mean square error of approximation (RMSEA), Tucker-Lewis index (TLI), and standardized root mean square residual (SRMR). We considered model fit to be adequate if the CFI and TLI values were higher than.90 and the RMSEA and SRMR values were lower than.08 [47]. Following the CFA, we assessed the reliability of the Polish DTCQ total score and subscale scores with Cronbach's alpha (α). We considered a reliability coefficient of.70 or higher as acceptable.

Next, we calculated linear correlations between the DTCQ total score and its subdimensions with the criterion variables, using Pearson's $r$ coefficient (for the MSPSS, we used Spearman's nonparametric test). Finally, we analyzed the associations between DTCQ scores and sociodemographic variables using independent t-tests for gender, Pearson correlations for age, and one-way ANOVAs with Bonferroni-corrected post-hoc tests for education level, relationship type, and caregiving reason, with appropriate effect size measures (Cohen's d and partial eta-squared).

### Results

We conducted a CFA to evaluate the factor structure of the Polish adaptation of the DTCQ. Two models were tested: a one-factor model and a two-factor model with correlated factors (see Table 2 and Supplemental Materials). Both models demonstrated acceptable fit according to conventional criteria [47], with RMSEA values below.05; CFI and TLI values above.90; and SRMR values below.08. The χ²/df ratios were all below 3, indicating an acceptable fit [52]. Factor loadings for the one-factor model ranged from 0.400 (item 5) to 0.715 (item 15), demonstrating moderate to strong loadings that indicate adequate representation of the underlying construct. The two-factor model showed similar loading patterns, with factor loadings ranging from 0.408 to 0.720 across both factors (see Supplementary Materials).

When single-factor and more complex solutions in model fit indices demonstrate similar and acceptable goodness-of-fit parameters, the principle of parsimony suggests selecting the simpler, single-factor model [51]. This approach tends

**Table 2. Fit Indices for the DTCQ models.**

| Model | χ²/*df* | RMSEA (90% CI) | CFI | TLI | BIC | SRMR |
|---|---|---|---|---|---|---|
| 1. One-factor | 2.45 | .049 (.041–.056) | .948 | .940 | 27461.33 | .036 |
| 2. Two correlated factors | 2.41 | .048 (.040–.055) | .950 | .942 | 27458.64 | .036 |

*Note.* The Supplemental Materials contain detailed tables with model coefficients, factor loadings, and graphical representations of all tested models (i.e., one-factor, two-factor, and hierarchical).

to yield more stable and replicable results across different samples, reducing the risk of overfitting and enhancing the model's generalizability [53]. Moreover, our CFA model revealed a much higher correlation between factors ($r = .96$) compared to the original paper ($r = .65$), which called into question the appropriateness of isolating two separate factors in this model, as postulated by the authors of this concept [17,19]. Given these findings, we retained the one-factor model for the Polish adaptation of the DTCQ. This model provides a good balance between statistical fit and theoretical parsimony while avoiding the interpretational challenges posed by the high-factor correlation in the two-factor model.

### Reliability

To assess the scale's internal consistency, Cronbach's alpha was calculated for the single-factor solution, with DTCQ scores computed as the mean value of 16 items, showing excellent reliability ($a = 0.91$).

### External validity of the DTCQ

To examine the relationships between dysfunctional thoughts about caregiving and various psychological and social factors, we conducted a correlational analysis, using the DTCQ general score. The results are presented in Table 3.

The analysis revealed a significant positive correlation between the DTCQ score and depressive symptoms. While the effect size was small according to Cohen's guidelines [54], this relationship suggests that higher levels of dysfunctional thoughts about caregiving were associated with more depressive symptomatology.

Regarding social support, as measured by the MSPSS, no significant correlations were found between the DTCQ score and overall perceived social support or support from specific sources (family, friends, and significant others). This indicates that dysfunctional thoughts about caregiving were not necessarily related to perceived social support in our sample. We also found significant positive correlations between the DTCQ score and both commitment and in-depth exploration, suggesting interdependencies between levels of dysfunctional thoughts about caregiving and identity commitment and in-depth exploration.

### Sociodemographic aspects of dysfunctional thoughts about caregiving

**Gender and age.** An independent samples *t*-test revealed a significant difference in DTCQ scores between genders (S1 Fig), $t(616) = 3.89$, $p < .001$, $d = 0.375$. Men (M = 2.03, SD = 0.672) scored significantly higher than women (M = 1.76, SD = 0.745). Age was positively correlated with DTCQ scores, $r(616) = .223$, $p < .001$, showing a small to medium effect size and indicating that older caregivers tended to report higher levels of dysfunctional thoughts about caregiving.

Table 3. Descriptive statistics and correlations of external measures with DTCQ general score.

| Variable | *M* | *SD* | *r* | *p* |
|---|---|---|---|---|
| SCL-27 Depression | 12.54 | 4.76 | .12 | .003 |
| MSPSS Total | 61.79 | 14.40 | -.01 | .310 |
| MSPSS Family | 20.13 | 5.97 | .03 | .476 |
| MSPSS Friends | 20.38 | 5.57 | -.04 | .350 |
| MSPSS Significant Others | 21.28 | 5.49 | -.01 | .902 |
| U-MICS Commitment | 14.58 | 4.36 | .24 | <.001 |
| U-MICS In-Depth Exploration | 18.10 | 3.37 | .09 | .023 |
| U-MICS Reconsideration of Commitment | 6.22 | 2.53 | -.04 | .308 |

*Note.* * $p < .05$, ** $p < .01$, *** $p < .001$. SCL-27 = Symptom Checklist-27 + Questionnaire; MSPSS = Multidimensional Scale of Perceived Social Support; U-MICS = Utrecht-Management of Identity Commitments Scale.

**Education.** A one-way ANOVA showed a significant effect of education on DTCQ scores, F(3, 614) = 10.4, $p < .001$, with a small to medium effect size ($\eta^2 p = .048$; [54]) (S2 Fig). Post-hoc comparisons with Bonferroni correction revealed that individuals with higher education scored significantly lower on the DTCQ compared to those with vocational education ($p < .001$) and secondary education ($p < .001$) (S2 Fig).

**Degree of kinship with the patient.** We observed a significant effect of the degree of kinship on DTCQ scores, F(5, 610) = 8.89, $p < .001$, with a medium effect size ($\eta^2 p = .068$; [54]) (S3 Fig). Post-hoc tests showed that caregivers of grandparents scored significantly lower compared to spouses ($p < .001$) and children ($p < .001$) of patients. Additionally, caregivers of parents scored lower than spouses ($p = .013$).

**Caregiving reason.** The reason for caregiving had a significant effect on DTCQ scores, F(3, 614) = 2.74, $p = .042$, though with a small effect size ($\eta^2 p = .013$; [54]) (S4 Fig). However, post-hoc comparisons did not reveal significant pairwise differences after Bonferroni correction. A trend suggested that caregivers of individuals with somatic diseases scored higher on the DTCQ than those caring for people due to old-age frailty ($p = .090$).

## Discussion

In the present study we evaluated the psychometric properties of the Polish version of the DTCQ and the links of dysfunctional thoughts with other caregiver characteristics. The obtained results per factor structure, internal consistency, and convergent validity suggest the DTCQ can be considered a valuable measure to assess dysfunctional thoughts among family caregivers of people with diverse chronic illnesses.

CFA indicated that both a one-factor and two-factor structure of the DTCQ fit the data well. Although the authors of the original version have suggested a two-factor solution, our results show that in Polish caregivers, a one-factor model is more suitable, reflecting the existence of a global predisposition towards dysfunctional thinking about caregiving. While the original study and its French version [17,19,27] found enough distinction to justify a two-factor structure, our results indicate that in the Polish cultural context, the perfectionism and responsibility dimensions were so highly interrelated that they may be better represented as a single, overarching construct. This difference might reflect cultural variations in how caregiving responsibilities and expectations are perceived and experienced in Poland compared to the original study's context, highlighting the importance of cultural adaptation in psychological measurement. The findings also indicate that the measure can be used with more diverse groups while maintaining its psychometric properties. This notion seems to be of great importance since the caregiving experience applies to people caring for those who need care for various reasons [55–56], and not only for people with dementia.

Our findings revealed positive relationships between DTCQ scores with identity commitment, showing that dysfunctional beliefs about caregiving are associated with a firm, stable identity, possibly also with a rigid self-image as a caregiver [17]. The observed positive correlations with in-depth exploration indicate that dysfunctional beliefs may co-occur with greater motivation to seek in-depth information about caregiving. Interestingly, no correlations were observed with reconsideration of commitment—an indicator of non-acceptance of the caregiver role. It appears that dysfunctional thoughts related to caregiving co-occurred with a strong, rigid, and well-defined caregiver identity rather than a diffuse identity and lack of acceptance of the caregiver role. This relationship can go in two different directions: It can be predicted that the dysfunctional thoughts associated with caregiving, such as the belief that only the individual can handle the task and that the carer should always be ready to care, support the emergence of such a well-defined identity, or on the other hand, a strong role identification may be a risk factor for experiencing care-related dysfunctional thoughts. Dysfunctional thoughts could have a double-edged sword effect, first helping to identify with the caregiving role and focus on caregiving tasks, but with long-term negative effects, by favoring maladaptive coping strategies such as reducing leisure time or inspiring less help-seeking [57]. Future research in this domain is needed to assess how dysfunctional thoughts may hinder or facilitate the process of adaptation and subsequent fulfilment of the role of a caregiver, and longitudinal studies might identify causal relationships between dysfunctional thoughts and caregiver identity development.

As expected, the DTCQ was linked to depressive symptoms. Our findings are mostly consistent with those of other studies (e.g., [17]) and the cognitive model [58–59], suggesting that dysfunctional thoughts are responsible for the development of depressive symptoms. The more rigid and self-demanding the perception of caregiving, the more strain and burden a caregiver may experience, leading to the feeling of overwhelm, a lack of hope and resources, and then eventually depression. Conversely, the severity of depressive symptoms may increase the intensity and frequency of dysfunctional thoughts [23], resulting in a greater risk of experiencing care-related negative beliefs due to a depressed mood before starting care.

While the observed relationships between dysfunctional thoughts and our study variables showed predominantly small to medium effect sizes according to Cohen's guidelines [54], these effects should not be underestimated in the context of caregiving research. Even small increases in dysfunctional thoughts may accumulate over time during the prolonged caregiving experience, potentially leading to meaningful changes in caregiver well-being and functioning.

Contrary to our expectations, we did not find links between the DTCQ and social support, similar to results obtained in previous studies [23]. Nevertheless, apart from the MSPSS subscale Family, the direction of these relationships remained consistent with the results of other studies [17,59]. It is also possible that in our Polish caregiver sample, factors other than social support were more related to dysfunctional thinking, such as the use of formal support resources (e.g., home care or day care). Overall, these findings provide insights into the complex relationships between dysfunctional thoughts about caregiving and various psychological and social factors, highlighting potential areas for further research and caregiver support interventions.

Regarding the sociodemographic characteristics, the proportion of female and male caregivers in our study confirms the results of previous studies indicating that women are more likely to take on the role of caregivers [60]. In contrast to our expectations, but similar to the results obtained by Montorio et al. [17], men reported more dysfunctional thoughts compared to women. This result confirms the latest research findings, indicating that both women and men experience negative consequences from caregiving, but perhaps in a different way [3,61]. Also, the differential effects between age and education level have precedents in other family care research [17,62]. It appears that older age and lower levels of education are risk factors for experiencing dysfunctional thoughts about caregiving.

Additionally, on closer examination, very interesting information came from comparing the intensity of dysfunctional thoughts between people who had different relationships with the patient. It appeared that the closer the social relationship was (e.g., child, spouse), the more dysfunctional thoughts about caregiving were present. These results are consistent with the results of another study that found that the severity of dysfunctional thoughts and psychopathological symptoms may depend on the degree of kinship and gender [63].

Finally, the results of our study suggest that the diagnosis of the family member is secondary to the severity of dysfunctional thoughts. Tandentik et al. [27] found differences in the intensity of dysfunctional thoughts between caregivers of patients with Alzheimer's disease and frontotemporal dementia. Nevertheless, these are disorders from the neurodegenerative spectrum, and to our knowledge, studies with more diverse groups have not been conducted. Therefore, perhaps it is not the disorder itself that is responsible for the negative experience of care but the coincidence of the cognitive profile and available resources in a cultural context that may be key for understanding the development and impact of dysfunctional thoughts [57,64]

## Study limitations

This study had some limitations. First, although we found no significant differences between the family member's diagnosis and dysfunctional thoughts, our sample was half-represented by caregivers of older people. Also, people with lower education were underrepresented, and the study was conducted online, thus the research was limited to internet users, and it is impossible to generalize these findings to general caregiving populations. Considering that our population of caregivers consisted also of older people that may not have had access to a computer, future research might replicate these



findings with a sample of Polish caregivers recruited through other means than online research. Future studies should seek more representative samples per educational levels regarding the caregiver sample and also include caregivers of people with somatic and mental health disorders in order to paint a fuller picture.

Another limitation pertained to the data collection method. We used self-report questionnaires that did not include scales measuring potential sources of bias in the answers provided (e.g., social desirability; [60]). Due to the sensitive topic of our study and perhaps the respondents' desire to appear in a better light, the results obtained may not fully reflect the severity of the variables. We recommend introducing scales that measure these features in future research.

Finally, we did not measure the convergent validity of the DTCQ with other measures of dysfunctional thoughts, as in Poland there is no alternative DTCQ questionnaire. Should an adaptation of another measure be published, it would be desirable to carry out such an analysis.

## Conclusion

The presented results suggest that the Polish version of the DTCQ is a reliable and valid measure for the assessment of caregivers' specific dysfunctional attitudes in various groups of caregivers. Considering an ageing society with a longer life expectancy and higher risks of health-related need for care, a tool like this might be very helpful for identifying people who find it difficult to adapt to the role of caregiver and to introduce adequate interventions for preventing or targeting psychological distress and other care-related vulnerabilities.

## Supporting information

**S1 File. Supplementary materials – DTCQ.**
(PDF)

**S2 File. DTCQ- polish version.**
(PDF)

**S3 File. DTCQ- english version.**
(PDF)

**S1 Fig. Mean DTCQ scores by gender. Note. Error bars represent 95% confidence intervals of the mean.**
(TIF)

**S2 Fig. Mean DTCQ scores by education. Note. Error bars represent 95% confidence intervals of the mean.**
(TIF)

**S3 Fig. Mean DTCQ Scores by degree of kinship with the patient. Note. Error bars represent 95% confidence intervals of the mean.**
(TIF)

**S4 Fig. Mean DTCQ scores by caregiving reasons note. Error bars represent 95% confidence intervals of the mean.**
(TIF)

## Author contributions

**Conceptualization:** Katarzyna Sanna, Konrad Piotrowski.

**Formal analysis:** Maciej Kościelniak.

**Funding acquisition:** Katarzyna Sanna, Konrad Piotrowski.



**Methodology:** Katarzyna Sanna, Maciej Kościelniak, Konrad Piotrowski.

**Project administration:** Katarzyna Sanna.

**Resources:** Katarzyna Sanna.

**Supervision:** Konrad Piotrowski.

**Writing – original draft:** Katarzyna Sanna, Konrad Piotrowski.

**Writing – review & editing:** Katarzyna Sanna, Maciej Kościelniak, Jarosław M Michałowski, Andrés Losada-Baltar, Konrad Piotrowski.

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
