## [Decision Letter · Decision Letter 0]

6 Jan 2025

PONE-D-24-53758Psychometric Properties of the Polish Version of the Dysfunctional Thoughts About Caregiving QuestionnairePLOS ONE

Dear Dr. Sanna,

Thank you for submitting your manuscript to PLOS ONE. After careful consideration, we feel that it has merit but does not fully meet PLOS ONE’s publication criteria as it currently stands. Therefore, we invite you to submit a revised version of the manuscript that addresses the points raised during the review process.

**Dear Authors,** **Thank you for your paper. I invite you to revise it according to the reviewers' comments. Please provide point-by-point responses whenever you agree or disagree with the recommendations. I am eagerly awaiting for the emended paper, and I wish you a sunny day.** **Handling Editor,****Paweł Larionow**

We look forward to receiving your revised manuscript.

Kind regards,

Paweł Larionow, Ph.D.

Academic Editor

PLOS ONE

“The authors Katarzyna Sanna, Konrad Piotrowski, Maciej Kościelniak, Jarosław Michałowski received a research grant as part of the 6th call for grant proposals for financing research projects at the Institute of Psychology (6 konkurs na finansowanie projektów badawczych w Instytucie Psychologii), SWPS University, Warsaw, Poland. Grant number - 1268-11”

4. We notice that your supplementary figures are uploaded with the file type 'Figure'. Please amend the file type to 'Supporting Information'. Please ensure that each Supporting Information file has a legend listed in the manuscript after the references list.

Reviewers' comments:

Reviewer's Responses to Questions

**Comments to the Author**

1. Is the manuscript technically sound, and do the data support the conclusions?

Reviewer #1: Yes

Reviewer #2: Yes

2. Has the statistical analysis been performed appropriately and rigorously? 

Reviewer #1: Yes

Reviewer #2: Yes

3. Have the authors made all data underlying the findings in their manuscript fully available?

Reviewer #1: Yes

Reviewer #2: Yes

4. Is the manuscript presented in an intelligible fashion and written in standard English?

Reviewer #1: Yes

Reviewer #2: Yes

5. Review Comments to the Author

Reviewer #1: I would like to express my gratitude to the PLOS ONE journal for the opportunity to review the article entitled “Psychometric properties of the Polish version of the Dysfunctional Thoughts about Caregiving Questionnaire.”

The article presents the adaptation of the Dysfunctional Thoughts About Caregiving Questionnaire into Polish, along with an analysis of its psychometric properties. It is well-structured and represents a significant and interesting contribution to the field. In my opinion, the article is suitable for publication after some modifications.

Below, I outline several suggestions for improving the manuscript, organized according to the sections of the paper.

Abstract

1. In line 30, I recommend replacing the term translation with adaptation, as this more accurately reflects the comprehensive scope of work undertaken, which extends beyond a straightforward translation.

Introduction

2. In the Introduction (lines 83–85), the authors mention various dysfunctional thought-related questionnaires used in caregiving research, identifying the Dysfunctional Thoughts About Caregiving Questionnaire (DTCQ) as one of these instruments. It would strengthen the manuscript to provide a clear justification for selecting this particular instrument for adaptation into Polish.

3. In lines 101–102, the authors state that the second objective of the study was to examine the relationships between the characteristics of family caregivers, contextual factors, and caregivers' quality of life. It would be helpful to clarify whether this objective is integrated into the analysis of the psychometric properties of the instrument and the search for evidence of validity, or if it represents an independent aim of the study.

4. In line 125, I suggest again replacing the term translation with adaptation for the reasons stated above.

5. In lines 132–135, the authors propose a hypothesis regarding the relationship between DTCQ scores, sense of identity in the caregiver role, and several contextual factors that influence caregiving. To enhance clarity, I recommend specifying the expected direction of these relationships.

Materials and Methods

Participants

6. It would be beneficial for the manuscript to specify the sampling procedure employed in the study.

7. In describing the participants' characteristics, some information is presented redundantly in both the text and Table 1 (e.g., gender and reasons for caregiving). I suggest removing the duplicate information from one of these locations. Additionally, within Table 1 itself, some rows contain repeated information, and the sample size is mentioned two times in the header. A thorough review of the table is recommended to address these issues.

Data Analysis

8. On lines 252–254, the authors should outline the procedures used to analyze the associations between DTCQ scores and sociodemographic variables, including the caregiver's age, education level, and relationship/role with the person needing care. I recommend also specifying the effect size indices associated with the relationships across all analyses performed.

Results and Discussion

9. In lines 277–279, the authors report that the internal consistency was calculated using Cronbach's Alpha, while in the Data Analysis section (lines 247–249), they state that McDonald's omega was used. This discrepancy should be resolved for consistency.

10. Effect sizes do not appear to have been adequately considered in the correlation analyses. For example, the correlation between the DTCQ general score and SCL-27 Depression, although statistically significant, is small. Similarly, the relationship with U-MICS In-Depth Exploration demonstrates a small effect size. It would be helpful to use and reference a guideline for interpreting the magnitude of such correlations.

This observation also applies to the effect sizes calculated in mean comparisons based on gender (Cohen’s d), where the effect size is small according to Cohen’s guidelines. Furthermore, I recommend calculating the corresponding effect sizes in pairwise comparisons of means within the ANOVA.

In general, all considerations regarding effect sizes should be reflected in the interpretation of the results and in the discussion derived from them.

I hope that the suggestions raised will be helpful to the authors in preparing another version of the manuscript.

Reviewer #2: Dear Authors,

Thank you for this paper.

The manuscript is well written. My suggestions are provided in the attached PDF file as comments.

In general, I suggest mediation analysis.

With best regards,

Reviewer

6. PLOS authors have the option to publish the peer review history of their article (what does this mean? ). If published, this will include your full peer review and any attached files.

**Do you want your identity to be public for this peer review?** For information about this choice, including consent withdrawal, please see our Privacy Policy .

Reviewer #1: No

Reviewer #2: **Yes: ** Ewelina Cichoń

---

## [Author Response · Author response to Decision Letter 1]

29 Jan 2025

Editor's comments

1) We have made changes according to guidelines

2) We added the following statement: "The funders had no role in study design, data collection and analysis, decision to publish, or preparation of the manuscript."

3) The data was made public on the OSF platform. The URL was attached to the submission: https://osf.io/r94q2/?view_only=4c3f174962674c2e8b4570e5c82c63d5

4) We amended the file type to 'Supporting Information" as recommended

Reviewers' Comments:

We have addressed all feedback from reviewers, a list of comments and our responses can be found in the document attached to the submission - Rebuttal letter

---

## [Editor Report · Decision Letter 1]

31 Jan 2025

PONE-D-24-53758R1Psychometric properties of the Polish version of the Dysfunctional Thoughts About Caregiving QuestionnairePLOS ONE

Dear Dr. Sanna,

Thank you for submitting your manuscript to PLOS ONE. After careful consideration, we feel that it has merit but does not fully meet PLOS ONE’s publication criteria as it currently stands. Therefore, we invite you to submit a revised version of the manuscript that addresses the points raised during the review process.

Dear Dr. Sanna,Thank you for your work on the revised version of the paper. I believe that all reviewers' comments were addressed properly, therefore, I do not strive to invite reviewers further. 1. Please indicate a sex/gender variable in Table 1. Please indicate an exact number of females, males, and if applicable non-binary people. Please do not treat this as a duplicate, as I would like to see these data in this table, not only in the main text, as the exact number of people from different gender groups is not mentioned.2. Procedure (lines 174-188). Please clarify what was the role of your professional Polish research and polling agency in data collection? What was the role of students who distributed the information of the study in their social media accounts? This is somewhat unclear. Please make this description very clear and transparent.3. Please delete "I" in lines 213-214. Here: "from 0 (I completely disagree) to 4 (I completely agree)."4. MSPSS: Please calculate internal consistency reliability for subscale scores. I find a Cronbach's alpha coefficient only for the total score. This is essential as subscale scores were used in this study. Please apply the idea of this comment to the U-MICS-IC. You can also indicate this reliability in the results section as this is a part of results.5. Please could you clarify whether you refer to your study results or to validation results published in other papers. Here and in similar cases/places: "The analysis was performed with AMOS 29 [46}. CFA had shown that the three-factor solution fit the data well, with CFI = .95, SRMR = .05, and RMSEA = .07 [47]. Cronbach's alpha was calculated, showing acceptable reliability (α = 0.75).". 6. Please indicate a specific date of IRB approval. Please indicate the name of institution.7. When comparing two CFA models, please consider using BIC as a criterion.8. Please prepare a supplementary table with item descriptive statistics (i.e., mean, standard deviation, skewness, kurtosis, min. and max. values).9. Please briefly mention factor loadings in the results section.10. I cannot find a table with descriptive statistics (i.e., mean and standard deviation) for all the study variables. Please incorporate these data into the table with correlation analysis. Please edit the paper according to these minor requests. I wish you an immaculate day!Paweł LarionowEditor

We look forward to receiving your revised manuscript.

Kind regards,

Paweł Larionow, Ph.D.

Academic Editor

PLOS ONE
---

## [Editor Report · Decision Letter 2]

26 Feb 2025

Psychometric properties of the Polish version of the Dysfunctional Thoughts About Caregiving Questionnaire

PONE-D-24-53758R2

Dear Dr. Sanna,

We’re pleased to inform you that your manuscript has been judged scientifically suitable for publication and will be formally accepted for publication once it meets all outstanding technical requirements.

Kind regards,

Paweł Larionow, Ph.D.

Academic Editor

PLOS ONE

Additional Editor Comments (optional):

Thank you for your revisions. I am accepting the paper. These are several comments for consideration:

1. I would suggest adding an abbreviation of the tool (DTCQ) in the end of the title.

2. I would recommend providing the name of the ethical committee in English, instead of using its Polish name, as the journal uses the English language.

3. I would suggest adding the deleted phrase "All items were assessed on a 5-point Likert scale, from 0 (I 216 completely disagree) to 4 (I completely agree).", when describing the DTCQ in the Measures section. When I was reading this phrase at the second round of revisions, I misunderstood this phrase, and incorrectly suggested deleting "I" in this phrase. As such, the phrase was correct. Therefore, if possible, please add it as it was.

4. Please include the Rebuttal letter #2 in the journal system as it seems this rebuttal letter is not present in the system.

Academic Editor

Paweł Larionow
---

## [Editor Report · Acceptance letter]

PONE-D-24-53758R2

PLOS ONE

Dear Dr. Sanna,

I'm pleased to inform you that your manuscript has been deemed suitable for publication in PLOS ONE. Congratulations! Your manuscript is now being handed over to our production team.

Kind regards,

on behalf of

Dr. Paweł Larionow

Academic Editor

PLOS ONE